# A Novel Mechanism for Bone Loss: Platelet Count Negatively Correlates with Bone Mineral Density via Megakaryocyte-Derived RANKL

**DOI:** 10.3390/ijms241512150

**Published:** 2023-07-29

**Authors:** Shohei Kikuchi, Akinori Wada, Yusuke Kamihara, Imari Yamamoto, Daiki Kirigaya, Kohei Kunimoto, Ryusuke Horaguchi, Takuma Fujihira, Yoshimi Nabe, Tomoki Minemura, Nam H. Dang, Tsutomu Sato

**Affiliations:** 1Department of Hematology, Faculty of Medicine, Academic Assembly, University of Toyama, 2630 Sugitani, Toyama 930-0194, Japan; skikuchi@med.u-toyama.ac.jp (S.K.); akino@med.u-toyama.ac.jp (A.W.); kamihara@med.u-toyama.ac.jp (Y.K.); s1950100@ems.u-toyama.ac.jp (I.Y.); odahamak@med.u-toyama.ac.jp (D.K.); kunimoto@med.u-toyama.ac.jp (K.K.); rhora@med.u-toyama.ac.jp (R.H.); tfujihi@med.u-toyama.ac.jp (T.F.); yoshimitb20@gmail.com (Y.N.); tominemura@gmail.com (T.M.); 2Division of Hematology/Oncology, University of Florida, Gainesville, FL 32610, USA; namdang5@outlook.com

**Keywords:** platelet, bone mineral density, megakaryocyte, RANKL

## Abstract

A potential association between hematopoietic stem cell status in bone marrow and surrounding bone tissue has been hypothesized, and some studies have investigated the link between blood count and bone mineral density (BMD), although their exact relationship remains controversial. Moreover, biological factors linking the two are largely unknown. In our present study, we found no clear association between platelet count and BMD in the female group, with aging having a very strong effect on BMD. On the other hand, a significant negative correlation was found between platelet count and BMD in the male group. As a potential mechanism, we examined whether megakaryocytes, the source of platelet production, secrete cytokines that regulate BMD, namely OPG, M-CSF, and RANKL. We detected the production of these cytokines by megakaryocytes derived from bone marrow mononuclear cells, and found that RANKL was negatively correlated with BMD. This finding suggests that RANKL production by megakaryocytes may mediate the negative correlation between platelet count and BMD. To our knowledge, this is the first report to analyze bone marrow cells as a mechanism for the association between blood count and BMD. Our study may provide new insights into the development and potential treatment of osteoporosis.

## 1. Introduction

Hematopoiesis is closely related to bones. The bone marrow fills small and irregular marrow cavities produced by mesh-like networks of trabeculae called spongy bones, which are surrounded by compact bones. The hematopoietic stem cells (HSCs) then attach to osteoblasts in the endosteum on the inside of the compact bone, and on the surface of the trabeculae, which is called the endosteal niche, a preferential site of residence for the most potent HSCs [1]. Therefore, it is possible that a rich bone mass supports robust hematopoiesis. Studies have focused on the relationship between the number of white blood cells (WBC), red blood cells (RBC), and platelets (Plt), which reflect hematopoiesis, and bone mineral density (BMD). However, published work on the possible correlation between BMD and WBC, RBC, and particularly Plt levels have demonstrated conflicting results.

Several reports have shown a negative correlation between Plt count and BMD [2,3,4]. Kristjansdottir HL, et al. conducted an analysis of 1005 men (median age 75.3 years, range 69–81) and found that the relationship between lumbar spine BMD (L1-L4) and Plt count was r = −0.06, *p* = 0.041, and that the relationship between total hip BMD and Plt count was r = −0.11, *p* = 0.003 [2]. Kim J, et al. examined 8634 subjects, including men over 50 years of age and postmenopausal women [4], and showed that in the highest Plt quartile, the odds ratio (95% confidence interval) of osteopenia vs. normal BMD was 1.39 (1.03–1.88), and that of osteoporosis vs. normal BMD was 1.60 (1.07–2.37). Furthermore, Li L, et al. analyzed 673 postmenopausal women and reported that the correlation between femur neck BMD and Plt count was r = −0.153, *p* = 0.001 [3]. Several hypotheses have been proposed for the potential mechanism responsible for the negative correlation between Plt count and BMD, including the involvement of serotonin [2], which is known to be negatively correlated with BMD [5], and is found in abundance in Plt [6].

On the other hand, there are published results that are not consistent with the findings above. Work by Valderrábano RJ, et al. [7,8] demonstrated no correlation between Plt count and BMD. The 2017 report included 2571 men aged 65 years and older [7], and the 2019 report included 5888 men and women also aged 65 years and older [8]. Meanwhile, other studies revealed a positive correlation between Plt count and BMD, with one by Schyrr F, et al. involving 143 patients prior to receiving chemotherapy for breast cancer [9], and another by Kim HL, et al. involving 338 postmenopausal women [10].

Thus far, there has therefore been no definitive conclusion on the association between Plt count and BMD, nor has there been any analysis of the possible biological mechanism responsible for any reported association. In this paper, we demonstrate a negative correlation between BMD and Plt count in men, with women being strongly affected by age factors. We propose that the underlying biological mechanism may involve the receptor activator of nuclear factor-kappa B ligand (RANKL), which is secreted by megakaryocytes responsible for the production of platelets.

## 2. Results

### 2.1. Patient Characteristics

Table 1 presents the clinical characteristics of the 65 patients in our study (33 females and 32 males). There was no difference in age between men and women. All patients have malignant lymphomas, the majority of whom have the histologic type diffuse large B cell lymphoma (DLBCL). The proportion of patients with advanced disease (clinical stage III/IV) compared to those with limited stage disease (clinical stage I/II) was higher in the female group than in the male group (*p* = 0.031). For BMD, the female group was clearly lower than the male group for L2-L4, L1-L4, total hip, and femur neck (*p* < 0.001). On the other hand, levels of WBC, neutrophils (Neu), lymphocytes (Lymph), RBC, hemoglobin (Hb), reticulocyte count (Ret), Plt, and tartrate-resistant acid phosphatase-5b (TRACP-5b), a marker of osteoclast number and bone resorption, showed no statistically significant difference between the sexes.

### 2.2. Correlation between BMD and Plt Count

As shown in Table 2, all four BMDs decreased with age in the female group. TRACP-5b level was negatively correlated with BMD in L2-L4, L1-L4, and total hip. For complete blood counts (CBC), positive correlations were observed between BMD and Lymph for L2-L4, L1-L4, and Total hip, but no correlation was observed between BMD and Plt. Results for the male group are presented in Table 3. BMD did not decrease with age; on the contrary, BMD in L2-L4 and L1-L4 increased with age. TRACP-5b level was negatively correlated with BMD in L1-L4 and total hip. Negative correlations were found between BMD and WBC and Neu in total hip and femur neck with respect to CBC. Negative correlations were also found between BMD and Plt for total hip and femur neck (both r = −0.365, *p* = 0.040). Since the negative correlation between BMD and Plt was confirmed only in male patients, the sample of female patients was not included in the subsequent analyses.

### 2.3. Correlation between BMD and Megakaryocyte-Produced RANKL

To evaluate for a potential biological mechanism involved in the observed negative correlation between Plt count and BMD, we focused on BMD regulators secreted by megakaryocytes (MGK), which are responsible for Plt production. Bone marrow mononuclear cells (BM-MNC) from male patients were induced to differentiate into MGK, and the production of RANKL, osteoprotegerin (OPG), and macrophage colony stimulating factor (M-CSF) were examined as BMD regulators (Table 4). MGK differentiation resulted in the enhanced expression of CD41a, a marker of MGK (Figure 1). Importantly, we observed a negative correlation between RANKL and BMD in the femur neck (r = −0.502, *p* = 0.034) (Table 4, Figure 2).

## 3. Discussion

By excluding women and focusing only on men in our present study, we were able to find a negative correlation between Plt count and BMD. It is our assumption that such a correlation between Plt count and BMD is also present in women; however, since aging has an extremely potent effect on BMD in women (Table 2), it may be difficult to delineate the relationship between BMD and other factors, independent of age. It is likely that a factor linking aging to lower BMD in women is a lower level of estrogen [11,12], with age-related estrogen withdrawal in women leading to decreased BMD through a variety of mechanisms [13]. 

In the female group in our study, there was a positive correlation between BMD and Lymph count (Table 2), which may also be due to the effect of estrogen. In a previous paper, it was reported that hormone replacement therapy with estrogen led to an increased lymphocyte ratio in postmenopausal women [14]. Therefore, it is possible that aging lowers estrogen levels, resulting in lymphocytopenia.

When considering the potential biological mechanism involved in the negative correlation between Plt count and BMD in the male group, we focused on MGK, given the dependence of Plt production on MGK. In addition, previous work [15] demonstrated that increased MGK resulted in greater MGK-derived RANKL production, associated with a higher level of osteoclasts in older mice than in younger mice. Since osteoclasts are responsible for BMD reduction, a factor linking increased Plt count with BMD reduction may be RANKL production by MGK. To evaluate this hypothesis, we examined the production of cytokines by MGK which can affect BMD, specifically RANKL, OPG, and M-CSF. RANKL and M-CSF induce osteoclast differentiation and decrease BMD, while OPG inhibits RANKL [16]. Our work revealed a negative correlation between RANKL production and BMD (Table 4), while the level of TRACP-5b, an activation marker of osteoclasts induced to differentiate by RANKL and others, was negatively correlated with total hip BMD (Table 3).

Besides Plt count, a negative correlation with BMD was observed for WBC in the male group, especially Neu count (Table 3). This relationship between Neu count and BMD in a male cohort (*n* = 2571) was also reported by another group [7]. Specifically, femur neck BMD was lower in men with higher Neu counts than in men with lower Neu counts (*p* = 0.016), and the decrease in total hip and femur neck BMD over time was greater (*p* < 0.001). This observed phenomenon may also be due to the involvement of RANKL, since a relationship between RANKL production in peripheral neutrophils and BMD loss has been reported in 59 male patients with chronic obstructive pulmonary disease (COPD) [17]. 

In our study, the inverse association between Plt count and BMD was revealed by excluding women who are greatly affected by aging, associated with a decrease in estrogen. Searching for a potential biological mechanism, we focused on MGK, the source of platelet production, and found that MGK production of RANKL was inversely correlated with BMD. This meant that the production of platelets and RANKL by MGK was responsible for the inverse correlation between Plt count and BMD. However, there are limitations to our conclusion. Specifically, we did not formally examine the MGK population itself, but rather the MGK-like cells induced to differentiate from BM-MNCs. In the present study, we utilized BM-MNC isolated and cryopreserved from residual bone marrow samples as a retrospective analysis. These BM-MNC were then induced to differentiate into MGK, since BM-MNC isolated by the standard Ficoll purification method do not contain MGK. In the future, we plan to isolate MGK directly from fresh bone marrow samples as part of a prospective study to further confirm our present conclusions.

Furthermore, it may be difficult to fully analyze the role of bone marrow-derived cells in studies of human BMD, since BMD measurements and bone marrow aspiration tests are typically never conducted in the same time frame. One exception would be the pretreatment systemic search for disease involvement performed on patients diagnosed with malignant lymphoma. Bone marrow aspiration tests as part of the pretreatment staging studies are done to evaluate for potential bone marrow infiltration of lymphoma cells, and pretreatment BMD measurements are obtained as baseline levels, out of concern for exacerbation of osteoporosis caused by the corticosteroids that are included in standard anti-lymphoma therapies, such as R-CHOP and others. 

However, these patients may have different characteristics from healthy individuals, or patients with other diseases that may affect BMD or Plt count. For example, malignant lymphoma may cause inflammation, which may alter the hematopoietic microenvironment and the cytokine profile. Moreover, our study is a correlation study, and does not prove a causal relationship between Plt count and BMD. It is possible that other factors may confound or mediate the association between Plt count and BMD, such as serum levels of hormones, vitamins, minerals, inflammatory markers, and Plt-derived factors. Further studies with larger and more diverse samples are needed to control for these potential confounders and mediators.

Additionally, further studies with functional manipulation of MGK or RANKL are warranted to elucidate the causal mechanism of Plt count on BMD. One of the future challenges is to perform gain-of-function and loss-of-function studies of RANKL production by MGK to confirm their effects on BMD. Such studies may involve genetic or pharmacological modulation of RANKL production by MGK in vitro or in vivo, and measurement of BMD changes using animal models or human subjects. Such studies may provide more convincing evidence for the role of Plt count in osteoporosis, and may lead to the development of novel therapeutic strategies. For instance, if RANKL production by MGK is confirmed to be a causal factor for BMD reduction, then targeting MGK may be a potential approach to prevent or treat osteoporosis.

## 4. Materials and Methods

### 4.1. Patient Samples

Patients with malignant lymphoma who had bone density tests (dual-energy X-ray absorptiometry; DEXA) using PRODIGY (GE HealthCare, Chicago, IL, USA) [18], blood tests for CBC and bone metabolism markers TRACP-5b, and bone marrow tests performed at the Department of Hematology, University of Toyama Hospital between April 2019 and November 2021 were included. This study was conducted according to the Declaration of Helsinki and was approved by the ethics committees of Toyama University Hospital (reference number R2021127). Written informed consent was obtained from all patients prior to study participation.

### 4.2. Cell Culture

The method for inducing differentiation of BM-MNC into MGK was previously described [19]. Lymphoprep^TM^ (Serumwerk Bernburg AG, Bernburg, Germany) was used to isolate BM-MNC from bone marrow puncture fluid. Isolated BM-MNC were cultured in MegaCult^TM^-C Medium with Cytokines (STEMCELL Technologies, Vancouver, BC, USA) for 14 days. It contains the following: Iscove’s Modified Dulbecco’s Medium, bovine serum albumin, recombinant human (rh) insulin, human transferrin (iron-saturated), 2-Mercaptoethanol, rh thrombopoietin, rh interleukin (IL)-6, rh IL-3, and supplements.

### 4.3. Flow Cytometry

For flow cytometric analyses, samples were collected using a FACSCanto II flow cytometer (BD Biosciences, San Jose, CA, USA) and analyzed with FlowJo software v9 (BD Biosciences). The expression of CD41a was analyzed using an FITC mouse anti-human CD41a, clone HIP8, antibody (BD Biosciences). 

### 4.4. Quantitative Real-Time RT-PCR

Following stimulation for MGK-differentiation, BM-MNC were lysed and total RNA was extracted using the RNeasy Plus Mini Kit (QIAGEN, Hilden, Germany) according to the manufacturer’s instructions. Total RNA (1 μg) was reverse transcribed using the QuantiTect Rev. Transcription Kit (QIAGEN). Quantification of mRNA was performed using the iQ5 Multicolor Real-Time PCR Detection System (Bio-Rad, Hercules, CA, USA) and the QuantiTect SYBR Green PCR Kit (QIAGEN). The obtained data were analyzed using iQ™5 Optical System Software, Version 2.1 (Bio-Rad), being normalized to glyceraldehyde-3-phosphate dehydrogenase (GAPDH) expression. The PCR primers were designed as shown in Appendix A. 

## 5. Conclusions

Osteoporosis is a common disease characterized by low bone mass and increased risk of fractures, especially in the elderly population. Although the etiology of osteoporosis is multifactorial, it is generally accepted that the balance between bone formation and resorption is crucial for maintaining bone health. Bone resorption is mainly mediated by osteoclasts, which are derived from hematopoietic stem cells in the bone marrow. Therefore, it is plausible that hematopoiesis and bone metabolism are closely related, and that changes in blood cell counts may reflect or affect bone mineral density.

In this study, we investigated the association between CBC and BMD in 65 patients with malignant lymphoma, who underwent both CBC and DXA measurements as part of their routine clinical assessment. We found that Plt count was negatively correlated with BMD, especially in the total hip and femur neck regions of middle-aged and elderly men. This finding was consistent with some previous studies that reported a negative correlation between Plt count and BMD in different populations [2,3,4]. However, other studies have shown conflicting results, such as no correlation [7,8] or a positive correlation [9,10] between Plt count and BMD. These discrepancies may be due to various factors, such as sample size, age range, sex distribution, menopausal status, comorbidities, medications, and measurement methods.

To explore the potential biological mechanism involved in the negative correlation between Plt count and BMD, we focused on MGK, which are responsible for Plt production in the bone marrow. MGK are known to secrete cytokines that can regulate BMD, such as RANKL, OPG, and M-CSF. RANKL and M-CSF stimulate osteoclast differentiation and activation, while OPG inhibits RANKL [16]. We hypothesized that an increased Plt count may reflect increased MGK activity, which may lead to increased RANKL/M-CSF or decreased OPG production and subsequent bone resorption.

To test this hypothesis, we induced BM-MNC from male patients to differentiate into MGK-like cells in vitro, and measured the production of RANKL, OPG, and M-CSF. Importantly, we observed a negative correlation between RANKL production and BMD in the femur neck (r = −0.502, *p* = 0.034). This finding suggests that RANKL production by MGK may mediate the negative correlation between Plt count and BMD. To our knowledge, this is the first report to analyze bone marrow cells as a mechanism for the association between blood cell count and BMD.

Our study provides new insights into RANKL production by MGK in osteoporosis and may have implications for the prevention and treatment of this disease. However, there are some limitations to our study. First, our sample size was relatively small and consisted of patients with malignant lymphoma, who may have different characteristics from healthy individuals, or patients with other diseases. Second, we did not directly measure the MGK population in the bone marrow, but rather used BM-MNC induced to differentiate into MGK-like cells in vitro. This may not fully reflect the actual MGK status in vivo. Third, we did not examine other factors that may influence BMD or Plt count, such as serum levels of hormones, vitamins, minerals, inflammatory markers, and Plt-derived factors. Therefore, further studies with larger and more diverse samples, direct MGK isolation and analysis, and comprehensive evaluation of other BMD-related factors are warranted to confirm and extend our findings.

## Figures and Tables

**Figure 1 ijms-24-12150-f001:**
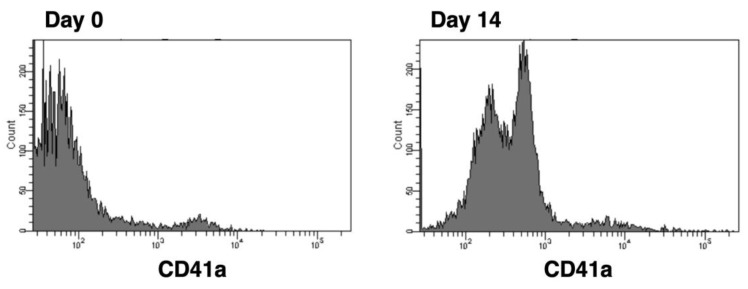
Megakaryocytic differentiation of BM-MNC. BM-MNC of patients were stimulated to differentiate into MGK. Megakaryocytic differentiation was estimated by the expression of CD41a using flow cytometric technique.

**Figure 2 ijms-24-12150-f002:**
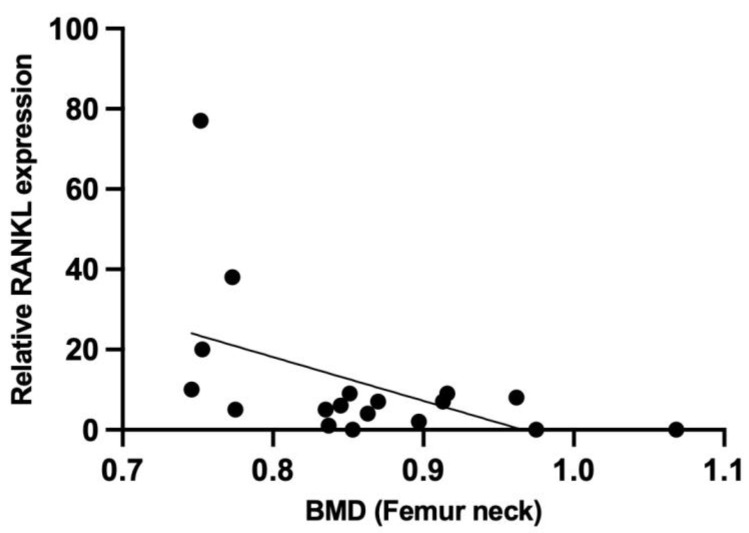
Correlation between BMD and RANKL-production of megakaryocytic-differentiated BM-MNC. Following stimulation of megakaryocytic-differentiation, production of RANKL in BM-MNC was estimated by quantitative real-time RT-PCR method, and was analyzed for correlation with femur neck BMD.

**Table 1 ijms-24-12150-t001:** Characteristics of the patients (*n* = 65).

	Female (*n* = 33)	Male (*n* = 32)	*p*
Age (years)	73 (53–87)	72 (39–84)	0.449
DLBCL/Others	21/12	25/7	0.199
CS I, II/III, IV	18/15	9/23	0.031
L2-L4 BMD (g/m^2^)	1.025 ± 0.147	1.258 ± 0.21	<0.001
L1-L4 BMD (g/m^2^)	0.99 ± 0.143	1.211 ± 0.197	<0.001
Total hip BMD (g/m^2^)	0.775 ± 0.115	0.914 ± 0.123	<0.001
Femur neck BMD (g/m^2^)	0.724 ± 0.113	0.852 ± 0.107	<0.001
WBC (×10^9^ L)	6.2 (2.5–16.5)	6.8 (3.6–12.0)	0.335
Neu (×10^9^ L)	3.7 (1.1–14.1)	4.3 (1.8–9.7)	0.503
Lymph (×10^9^ L)	0.9 (0.3–1.6)	0.9 (0.2–1.7)	0.870
RBC (×10^12^ L)	4.2 (2.4–5.0)	4.1 (2.8–5.4)	0.707
Hb (g/L)	122 (80–166)	125 (81–156)	0.798
Ret (×10^9^ L)	54 (29–150)	59 (33–130)	0.772
Plt (×10^9^ L)	246 (164–492)	258 (68–420)	0.644
TRACP-5b (mU/dL)	419 ± 152	410 ± 166	0.827

**Table 2 ijms-24-12150-t002:** Spearman analysis between BMD and selected variables in female patients (*n* = 33).

BMD	L2-L4		L1-L4		Total Hip		Femur Neck	
	r	*p*	r	*p*	r	*p*	r	*p*
Age	−0.283	0.022	−0.288	0.020	−0.491	<0.001	−0.576	<0.001
WBC	0.017	0.892	0.037	0.771	−0.012	0.927	−0.172	0.170
Neu	−0.087	0.492	−0.049	0.697	−0.126	0.316	−0.204	0.103
Lymph	0.257	0.039	0.254	0.041	0.263	0.034	0.040	0.752
RBC	0.046	0.715	0.004	0.975	−0.021	0.866	−0.076	0.545
Hb	−0.079	0.531	−0.129	0.306	−0.014	0.914	0.001	0.996
Ret	0.100	0.429	0.064	0.614	0.115	0.362	−0.065	0.606
Plt	0.169	0.354	0.165	0.376	−0.050	0.784	−0.079	0.667
TRACP-5b	−0.302	0.015	−0.302	0.014	−0.292	0.018	−0.022	0.862

**Table 3 ijms-24-12150-t003:** Spearman analysis between BMD and selected variables in male patients (*n* = 32).

BMD	L2-L4		L1-L4		TOTAL HIP		Femur Neck	
	r	*p*	r	*p*	r	*p*	r	*p*
Age	0.310	0.012	0.314	0.011	0.111	0.381	−0.047	0.709
WBC	−0.188	0.134	−0.208	0.096	−0.314	0.011	−0.324	0.008
Neu	−0.215	0.086	−0.236	0.059	−0.374	0.002	−0.328	0.008
Lymph	0.079	0.532	0.048	0.706	−0.009	0.941	−0.063	0.618
RBC	−0.168	0.182	−0.195	0.120	−0.055	0.666	−0.049	0.699
Hb	−0.053	0.673	−0.072	0.567	0.123	0.330	0.091	0.471
Ret	0.220	0.078	0.212	0.091	0.132	0.293	0.222	0.075
Plt	−0.254	0.160	−0.284	0.115	−0.365	0.040	−0.365	0.040
TRACP5b	−0.216	0.084	−0.250	0.045	−0.312	0.011	−0.214	0.087

**Table 4 ijms-24-12150-t004:** Spearman analysis between BMD and key OC factors in MGK of male patients (*n* = 18).

BMD	L2-L4		L1-L4		Total Hip		Femur Neck	
	r	*p*	r	*p*	r	*p*	r	*p*
RANKL	−0.266	0.286	−0.330	0.181	−0.399	0.101	−0.502	0.034
OPG	−0.108	0.670	−0.138	0.586	−0.216	0.390	−0.425	0.079
M-CSF	−0.010	0.968	−0.013	0.958	−0.124	0.623	−0.056	0.827

## Data Availability

Data are available on reasonable request. All data relevant to the study are included in the article.

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
