# Peer review of "A Novel Mechanism for Bone Loss: Platelet Count Negatively Correlates with Bone Mineral Density via Megakaryocyte-Derived RANKL"

_ijms, 2023, doi:10.3390/ijms241512150_

Round 1

Reviewer 1 Report

Biological factors that link blood count and bone mineral density are still unknown. The authors show for a first time the analysis of the bone marrow cells as a mechanism for the association between blood count and bone marrow density.

1. I suggest to move the gene sequences from section 4.4  to Supplementary
2. Please enter more details in the section Conclusions.
3.Please, check if all abbreviations are explained if occur for a first time.
4. Please standardize text throughout manuscript.

Author Response

Author's Reply to the Review Report (Reviewer 1)

Comments and Suggestions for Authors

Biological factors that link blood count and bone mineral density are still unknown. The authors show for a first time the analysis of the bone marrow cells as a mechanism for the association between blood count and bone marrow density.

  1. I suggest to move the gene sequences from section 4.4 to Supplementary.

We have done as instructed. Lines 265, and 310-311.

  1. Please enter more details in the section Conclusions.

We have done as instructed. Lines 267-309.

  1. Please, check if all abbreviations are explained if occur for a first time.

We have checked all the abbreviations. Lines 81, 87, and 102.

  1. Please standardize text throughout manuscript.

We have reviewed the “Instructions for Authors” again and made corrections regarding the abbreviations as you pointed out.

Reviewer 2 Report

The authors of the manuscript entitled  " A Novel Mechanism for bone loss: Platelet count negatively correlated with bone Mineral density vis megakaryocytic-derived RANKL" have shown in this manuscript a potential association between blood platelet count and bone mineral density. the authors have teased out the negative correlation of the platelet count and bone mineral density in the male groups. The female group has been excluded due to other factors such as aging that might play a role in the loss of bone mineral density simply to to hormonal changes. the authors in their experiments look at megakaryocytes for platelet production that secrete cytokines that regulate bone mineral density such as OPG, M-CSF and RANKL. In this they also found a negative correlation between RANKL production by megakaryocytes-platelets and bone mineral density in males. While this marks only a correlation study between the platelet count and the bone mineral density, the authors should mention the same in their results and conclusions. I feel the authors would increase the potential of this paper if it is supported by gain of function and loss of function studies to drive home their point.

In addition to that I only have minor comments regarding the manuscript.

a) the authors when using abbreviations should expand on it the first time they use it, such as CBC.

b) with regards to the Figure1 and Figure 2, the authors should stick to a common font size.

c) the conclusions seems very redundant and the authors needs to expand on their conclusion.

Author Response

Author's Reply to the Review Report (Reviewer 2)

Comments and Suggestions for Authors

The authors of the manuscript entitled "A Novel Mechanism for bone loss: Platelet count negatively correlated with bone Mineral density vis megakaryocytic-derived RANKL" have shown in this manuscript a potential association between blood platelet count and bone mineral density. The authors have teased out the negative correlation of the platelet count and bone mineral density in the male groups. The female group has been excluded due to other factors such as aging that might play a role in the loss of bone mineral density simply to hormonal changes. The authors in their experiments look at megakaryocytes for platelet production that secrete cytokines that regulate bone mineral density such as OPG, M-CSF and RANKL. In this they also found a negative correlation between RANKL production by megakaryocytes-platelets and bone mineral density in males. While this marks only a correlation study between the platelet count and the bone mineral density, the authors should mention the same in their results and conclusions. I feel the authors would increase the potential of this paper if it is supported by gain of function and loss of function studies to drive home their point.

Thank you for your valuable comments. We have added the content you suggested to Discussion. Lines 212-230.

In addition to that I only have minor comments regarding the manuscript.

a) The authors when using abbreviations should expand on it the first time they use it, such as CBC.

We have checked all the abbreviations. Lines 81, 87, and 102.

b) With regards to the Figure1 and Figure 2, the authors should stick to a common font size.

We have followed your instructions and reduced the font size of Figure 1 and 2.

c) The conclusions seem very redundant and the authors needs to expand on their conclusion.

We have done as instructed. Lines 267-309.